# From Reporting to Removing Barriers: Toward Transforming Accommodation Culture into Equity Culture

Alison Cook-Sather [1],* and Morgan Cook-Sather [2]

1   Education Department, Bryn Mawr College, Bryn Mawr, PA 19010, USA
2   Class of 2025, The College of William & Mary, Williamsburg, VA 23185, USA; mecooksather@wm.edu
*   Correspondence: acooksat@brynmawr.edu

**Abstract:** This reflective case study is situated at the intersection of the literature on pedagogical partnership, child-parent research, and Critical Disability Studies. It presents a mother/daughter, faculty/student exploration of the daughter's lived experiences of navigating, as a legally blind person, the campus and courses of a college designed for fully sighted students. After presenting our conceptual frameworks and describing, using text and a video, the daughter's lived experience of navigating the accommodation culture on her campus, we describe the semester-long partnership process through which the video was created with the goal of moving faculty, staff, and students toward equity culture. To support others in developing such video projects on their own campuses, we draw on details of this partnership to offer guidelines for co-creating representations of the lived experiences of other students with disabilities. By synthesizing learnings from this experience and the literature noted above, we offer recommendations for transforming accommodation culture into equity culture. These recommendations include: establishing diversity as the norm in every learning context; intentionally inviting a revision of differences from deficits to resources; going beyond providing accommodations to understand students' lived experiences; and sharing the active taking of responsibility for shifting from accommodation to equity culture.

**Keywords:** student-faculty partnership; child-parent research; disability; accommodations; barrier-free design; equity

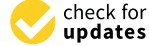



## 1. Introduction

Both legislation and practice have expanded access in higher education in the United States for students with disabilities. The Americans with Disabilities Act (ADA), passed in 1990, prohibits discrimination against and requires reasonable accommodations for people with disabilities [1]. However, the disability services offices on college campuses that have been established to meet both legal and ethical obligations, and to offer guidelines for creating flexible and responsive learning environments according to principles of Universal Design (UD), provide uneven support for disabled students across US institutions of higher education, with a high percentage of faculty both unaware of and resistant to UD practices in particular [2]. These realities create a culture—a set of values and customs—premised on (often grudging and inadequate) accommodation rather than equity.

Challenging this premise and set of practices, a growing number of scholars have argued that it is terms such as 'disabilities' and 'normal' and the culture created through their use, instead of the conditions of the people themselves, that disable [3–6]. McDermott and Varenne succinctly capture this argument: "A disability may be a better display board for the weaknesses of a cultural system than it is an account of real persons" [7]. In a cultural system premised on accommodation, the default is adapting to meet a single, unquestioned, and exclusive norm. In a cultural system premised on equity, differences are valued as resources, and customs emerge through the co-creation of inclusive conditions under which all can thrive.

We report here on a project that has the potential to contribute to a shift from accommodation to equity culture in higher education. This reflective case study is co-authored by Alison Cook-Sather, a professor of education and international leader in student-faculty pedagogical partnership, and Morgan Cook-Sather, a second-year undergraduate at The College of William & Mary (W&M) with a condition that renders her legally blind and who has found ways to succeed in school despite the barriers she has encountered. Enacting a mother-daughter as well as a faculty-student pedagogical partnership, our work draws on the various kinds of advocacy and agency that we have found necessary to navigate educational contexts whose cultural systems assume, design for, and privilege full sightedness, even as they extend access to students with a variety of disabilities, including forms of blindness.

Understandings of access in higher education typically focus on offering approaches for entry into an institution, classroom, activity, or assignment. They do not often consider what students experience after entry has been gained. Our title references the stance toward promoting access that W&M takes: "Report a Barrier." This stance sends the encouraging message that it is the institution's responsibility to remove barriers. However, the reality is that, in the still-dominant accommodation culture that centers norms to which approximation is the goal, students themselves, including Morgan, must do much of the work of that removal while the culture remains unchanged. With an expanding community of scholars, practitioners, and learners, we argue for taking a step beyond removing barriers.

In the following section, "Intersecting Conceptual Frameworks," we situate our work in pedagogical partnership, child-parent research, and Critical Disability Studies, all of which affirm calls to illuminate, challenge, and redress the harm that equity-denied students experience in higher education [8]. In the section called "Our Experiences of Reporting Barriers within an Accommodation Culture," we use text and a video to capture Morgan's lived experiences of navigating the W&M campus and her courses. In the subsequent section, "Guidelines for Representing Students' Lived Experiences," we draw on the details of Morgan's particular co-created video project to offer guidelines to others for developing such video projects on their own campuses. In the final section, "Recommendations for Transforming Accommodation into Equity Culture," we synthesize learnings from Morgan's experience to suggest how this work can move higher education toward equity culture. We conclude by linking back to key arguments from the literature we have cited and presenting our hopes for the future.

## 2. Intersecting Conceptual Frameworks

Pedagogical partnership, child-parent research, and Critical Disability Studies share a focus on lived experiences in context, as well as the necessity of drawing on more than one perspective in order to develop spaces for and practices of learning for a diversity of people. All three call into question standard assumptions about how living and, in particular, learning environments might be conceptualized and constructed. In addition, all three focus on creating inclusive and equitable spaces and practices.

Pedagogical partnership in higher education contexts is commonly defined as "a collaborative, reciprocal process" through which all participants "have the opportunity to contribute equally, although not necessarily in the same ways, to curricular or pedagogical conceptualization, decision making, implementation, investigation, or analysis" [9]. By deliberately positioning students "alongside others with educational expertise" [10], pedagogical partnership affords faculty and staff the opportunity to "engage students as co-learners, co-researchers, co-inquirers, co-developers, and co-designers" [11]. When students take on liminal roles—positioned in a kind of in-between space: not enrolled students, but informed as students with particular lived experiences and expertise [12]—they can mobilize their own identities to lead change for equity and inclusion [13]. In relation to disability in particular, rather than accepting individuals as "deficient" and in need of

service interventions, a partnership approach "draws on disabled students' expertise to address barriers within the institutional environment" [14].

Child-parent research recognizes youth as researchers with the capacity to engage in collaborative investigations with other members of their families [15]. As a mother-faculty member partnering with a daughter-student, we draw on our respective sources of expertise—contributing equally, although not necessarily in the same ways [9]—to analyze Morgan's lived experiences of her college and of her partnership in creating a video to share those lived experiences. We recognize that different educational, geographical, and political contexts, as well as individual parents' socio-economic conditions and socio-cultural identities, afford parents a greater or lesser opportunity to engage in such partnership work and research. These realities are areas for further critical exploration and transformation.

Calling for a "reflexive and politicised Critical Disability Studies," Goodley and colleagues argue that "to contemplate disability is to scrutinise inequality," and that the field of disability studies has produced a body of theoretical work "that is, broadly speaking, counter-hegemonic to dominant understandings of disability" [5]. The work that has been done in disability studies on the built environment offers useful language, sometimes literal and sometimes metaphorical, to describe barriers to access and therefore to equity. Hamraie notes that the work of activists in the 1960s and 1970s "resulted in the passage of federal civil rights legislation that aspired to protect the access of people with disabilities to the built environment," and the architectural strategies that underlie these legislative gains came to be known as "barrier-free design" [6]. Echoing McDermott and Varenne's point regarding disabilities being better display boards "for the weaknesses of a cultural system" than accounts of real persons [7]. Hamraie notes that feminist philosophies of disability "argue that disability is a product of the built and social environments, rather than a medical state that is intrinsic to the body of a given individual" [6].

The design of buildings, Hamraie notes, "is not a value-neutral and passive act; rather, the design of the built environment actively conditions and shapes the assumptions that the designers, architects, and planners of these value-laden contexts hold with respect to who will (and should) inhabit the world" [6]. The design of classrooms and associated assignments is also not neutral and reflects certain assumptions about the expected, acceptable, and possible ways of engaging. As Brown and colleagues argue: "Similar to built physical environments, websites are often designed by non-disabled people with non-disabled users in mind" [14], and yet accessing and navigating websites is increasingly expected in college courses. The design of almost all aspects of most courses, actual and virtual, is referenced to and undertaken by the able for the able, without consideration of barriers that might dis-able.

Embracing a "barrier-free design" in classrooms certainly includes considerations of physical space, but it also requires considerations of the ways in which students are invited and expected to engage in all aspects of work for the course—in person and online. We argue not only for consideration of the learning space, but also of the ways of engaging within and beyond it, specifically for those whose norm does not match the norm for which the space was built and the modes of engagement imagined. As Dolmage asserts, all of us in higher education are "involved in the continued production of space, . . . [and] . . . students should be agents in this negotiation" [16]. Linking to this point about agency, Schley and Marchetti provide an example of shifting from "the traditional accommodation mindset" through which staff "provide the accommodations required by law" and "instructors and other students are left with little agency or responsibility beyond providing for the specified accommodations to take place" [17]. Through what Schley and Marchetti call "the inclusion mindset," instead of belonging to "a third-party disability services office," responsibility and agency are "re-centered in the classroom" [17].

### 3. Our Experiences of Reporting Barriers within an Accommodation Culture

As a mother-faculty member and a daughter-student pair, we have different angles on the experiences of reporting barriers within an accommodation culture and are positioned

differently to try to move higher education beyond that norm. In this section, we write in our own voices to share these different angles. We focus primarily on Morgan's experience, with Alison's support intended to legitimize that experience.

Morgan: I have been navigating cultures that assume and spaces that are built for fully sighted people all my life, including when I had full sight up until age nine. At that point, I became legally blind due to the onset of a form of juvenile macular degeneration. I have peripheral vision, so I can navigate literal spaces. Because my central vision is gone, though, regular (or even significantly enlarged) text is illegible to me, as is any written or printed text on black- or whiteboards, projected slides, or other visuals in classrooms, no matter how close to the front of the room I sit. It also takes me a very long time to navigate resources in an electronic form; I must constantly toggle between enlarging the text on the computer screen so I can read it (often as large as 72-point font), searching for the particular text or image I need to take in, and navigating to the spaces that need to be filled in or otherwise engaged with. In a short video called "Navigating Campus: An Accessibility Story" [18], produced though a partnership between me and W&M's Studio for Teaching and Learning Innovation (STLI), I discuss and we show what navigating the campus, classrooms, and electronic texts is like for me.

At the urging of my mother and then on my own initiative, I have advocated for myself since late elementary school, meeting with my teachers each semester to inform them of my access needs and finding ways to teach myself in new ways (see my high school senior project on inventing a new way to learn piano pieces [19]). While most of my teachers have been very responsive, all of my work to gain access has unfolded within an accommodation culture. I have had to use the language specified by legislation, which is often pathologizing, and I have been accustomed to feeling gratitude for accommodations that make it possible for me to do the work, but do not create equitable conditions for doing so.

In preparation for my arrival at W&M, I completed the process of registering with Student Accessibility Services (SAS) and detailing the accommodations I would need. While the spirit behind W&M's "Report a Barrier" stance is a huge improvement over messaging that pins the responsibility on students to surmount barriers, my experience illustrates how much of the work of gaining access still falls to students. I narrated this experience in response to a study abroad application essay prompt that asked me to share a time that I dealt with an academic challenge and what steps I took to overcome this challenge. I expand on what I wrote in that essay below.

After undergoing the extensive intake process through W&M's SAS and receiving a formal letter of accommodations that were to be implemented, I believed my needs would be met. However, my approved accommodation—the allowed use of a digital format for exams—has often not been available when I arrive at the SAS office to take my exams. I requested this accommodation because I have completed all my assignments digitally for years; it is only in this format that I can manipulate the screen to ensure that the material is large enough for me to read.

I take my exams in the SAS testing center. During my first year at W&M, I was given a paper copy every time. Each time I explained my accommodation to the proctor, and they went off to sort it out, leaving me moments before the exam, when I was already nervous about the material, to think about how I was going to get access to the exam itself. By the time I received the correct format, I had lost mental capacity that I needed for the actual exam. I emailed SAS to ask what I could do to clear up this miscommunication, and after a constructive conversation, I figured that everything would run smoothly from there on out.

Unfortunately, during the fall of my second year at W&M, I was again greeted with a paper copy of an exam. I again informed the proctor of my situation, and it was 30 min before I received the version I needed. After that exam, SAS reached out, apologizing for the miscommunication, and asked me if I wanted to use a magnifier instead. I responded by reiterating why the accommodation I already had suited me best. They informed me that professors had requested that, unless under special circumstances, exams are not to be

administered digitally. I held my ground, insisting that my situation warranted this option and explaining specifically why the digital format works so well for me. I even offered to meet with them to demonstrate my experience so they could know what it is like for me to engage with materials. The digital version does not place me on a level playing field with other students; once I have an accessible version, there is still a great deal of additional work involved in reading the material.

Up until this that last email exchange, I had felt extensive frustration, paired with additional stress and anxiety. However, when I received that last email, I did not let it affect me, and focused on arguing my case using intentional language to convey the psychological effects this situation has had on me, in addition to the physical manifestation of my visual impairment and its severe impact on my learning. Utilizing a particular language to describe my experience to people who do not understand its complexity is a skill I have honed throughout this process. Accepting the reality that I have to do this extra work and choosing to focus on how best to approach it rather than focusing on the fact that I have to do it at all allows me to not further hinder my experience, but maintain my power. Growing comfortable with being uncomfortable, developing coping mechanisms for these situations, and developing a greater capacity to respond to situations that are beyond my control have better equipped me for overcoming obstacles in the future.

Alison: Supporting Morgan in advocating for herself has made me feel equally proud of and inspired by her, but increasingly frustrated with and sometimes deeply angry at the systems and practices that continue to normalize barriers and perpetuate inequities. While Morgan's tenacity and capacity to advocate for herself might be seen as laudable, this story (a version of which so many students could share) is one of having to take on the work that the system promises to do—remove a barrier—on top of the already extra work she has to do when engaging with the course materials after gaining access to them. Her disability is, to evoke McDermott and Varenne once again, a display board for the weaknesses of the cultural system [7]. Arguably, Morgan is more able than students who experience no barriers; because the cultural system is designed for them, they can focus all their energy on the work at hand, whereas Morgan has to expend time and energy gaining access to the materials, developing additional skills as well as focusing on the work. As a parent and an educator, this reality is doubly painful to me.

As a believer in and facilitator of partnership as an approach to redressing inequities [8], I arranged for Morgan and me to meet with the assistant director of the STLI at W&M and two student partners who work in the STLI. Over Zoom, we explored the ways in which Morgan's experience might be a good fit for the STLI's approach to crafting and sharing stories of students' learning experiences to inform teaching practices at W&M. All agreed that it was indeed a good fit, so in the early weeks of the Spring-2023 semester, Morgan worked closely with one of the STLI student partners, Ryan Harp, to plan their approach and create a video. In the section below, we enumerate the steps Morgan and Ryan, and another student partner, Jada Bonds, took in the form of guidelines for a partnership approach to creating a representation of a student's lived experience that can contribute to the shift from accommodation culture to equity culture.

## 4. Guidelines for Co-Creating Representations of Students' Lived Experiences

We drafted and revised this set of steps throughout Morgan's experience with W&M's STLI. While we draw on the details of Morgan's particular partnership to offer these guidelines, we recognize that contexts, support structures, and individual experiences intersect in unique ways in every case, and so these should, of course, be adapted as needed.

### 4.1. Consider Who Should Be Involved in the Project

Teaching and learning centers, and pedagogical partnership programs in particular, are obvious partners for this kind of work. A director, co-director, or other staff member might initiate the project, or a student can advocate for themselves, as Morgan did with

Alison's support. In addition, as with much pedagogical partnership work, the project might be led by one or more student partners, as it was by Ryan in Morgan's case.

Consider which student partners in the teaching and learning center/partnership program would be best positioned to be involved and why. For instance, Film Studies students, such as Ryan, are especially well positioned to draw on their knowledge and skills in making videos, and they could be hired as student partners if they are not in those roles already. In addition, consider which other students might be interested in such a project and why. Drawing in other students can build capacity and community and also open up additional opportunities for collaboration.

Finally, in addition to teaching and learning center and/or partnership program student partners, consider which members of other offices on campus might be good partners, such as staff from access services, multimedia services, libraries, or instructional technology. These are important partners to seek out both for the expertise they bring and also out of respect for their offices' official responsibilities.

### *4.2. Convene an Initial Planning Meeting*

Convene a meeting of the director of the teaching and learning center/partnership program and the selected student partners in that program, the student whose story will be featured, and, where appropriate, other advocates, allies, or collaborators, such as those mentioned above. Invite everyone at the initial meeting to share their thoughts on what the focus of the story should be: what the lived experience is, what people tend not to understand about it, what is most important to capture and convey with the story, etc.

The initial meeting we had with the assistant director of W&M's STLI, Ryan, and Jada was extremely generative in the ways in which it allowed us all to find language to name Morgan's experience, identify what it takes to convey that experience to others who do not share it, and determine what work such naming and sharing can accomplish in moving from accommodation to equity culture. Every one of us in the conversation said at one point or another some version of, "I hadn't thought of that," suggesting that the intersection of our multiple perspectives—the premise of partnership—was key to our conceptualizing this project. In particular, Jada noted the potential power of this work for other students, not just faculty and staff—a point to which we will return later.

### *4.3. Develop a Set of Steps to Enact the Plan*

With the participants and goal in place, take a series of steps that include generating guiding questions, deciding what can be told and what needs to be shown, selecting faculty and staff members who can share their experiences of working with the focal student, and engaging in an iterative process of production.

### 4.3.1. Generate Guiding Questions

Formulating questions requires identifying and naming what is most essential to understand about any given phenomenon. It requires a deeper analytical process than straight narrative. The process of generating and revising questions to be addressed prepares participants to engage in a dialogue—a more dynamic mode of exploration and representation than a monologue or other one-sided representations, which can objectify.

Ask the student partners from the center and the student whose story will be told to generate questions that the featured student will address in a videotaped interview. This ensures that both those producing the story and those whose story is being produced contribute to its shaping. Again, this draws on the partnership principle of multiple perspectives being brought to bear, thereby "contributing equally, although not necessarily in the same ways" [9] to the partnership.

### 4.3.2. Decide What Can Be Told and What Needs to Be Shown

Some things can be easily grasped through description, and some require simulated experience to grasp. Decide what can be described using words and what needs to be

simulated visually or aurally for those who do not share the focal experience. For instance, the fact that Morgan cannot see what is written or printed on black- or whiteboards, projected slides, or other visuals in classrooms is a reality that most people can comprehend if it is simply explained. However, the experience of navigating the campus or an electronic version of a document is not; until people see what Morgan experiences and has to do, they cannot imagine or understand. Therefore, conveying those aspects of her story requires simulation. The video Ryan created captures Morgan's daily experiences of mapping routes across the W&M campus to avoid stairs and using hearing rather than facial recognition to identify people, as well as seeing total blurs on slides or whiteboards no matter where in the classroom she sits.

### 4.3.3. Include Faculty Members' Insights

Faculty are often most quickly convinced by other faculty. Hearing a colleague describe the insight they gained and the ways in which they have revised their approaches to creating a classroom environment or structuring an assignment can facilitate the shift from accommodation to equity culture. Therefore, ask the focal student to identify faculty members who have had the student in their classes, have come to understand the student's lived experience, and can speak about the understanding they have gained of that experience. In Morgan's case, she conveyed those faculty members' understanding and actions in her video. Actions faculty members took as a result of their understanding included reading aloud and audiotaping chapters that did not come in any accessible form and changing approaches to coursework after they saw what Morgan had to do to navigate materials on her computer screen.

### 4.3.4. Engage in an Iterative Process of Production and Checking in

It is important for the people who live the story and the people who produce the story to stay in conversation. The person narrating has the lived experience and knowledge to share; the person who has the resources to capture and present the narrative knows videography and how to capture stories. While, after following the steps above, the people who are creating the video have an idea of what they want to convey, it is essential to ensure that they keep in mind how the person whose story is being featured wants it to be told. If producers run the show, they might distort or not quite communicate what the person telling the story wants to convey. Thus, it is important to stay in conversation along the way and to ensure that ideas are aligning and that both parties are happy with the presentation. This must be a process of mutual respect between both sources of expertise: the person who has the experience and the person who produces the video.

Here are the steps that Morgan and Ryan took (with Jada also involved in the early steps):

- Complete all of the steps listed in the sections above;
- Create a schedule for the student partner(s) and other support staff to:
  - conduct and videotape interviews with the focal student using both sets of questions, interviews with identified faculty, and other clips of the focal student's experience;
  - film segments to be integrated into the video;
- Student partner(s) (and other support staff?) from the center create a short video;
- Everyone involved confers to make sure they are all OK with the excerpts from the interview to be included in the video;
- Student partner(s) (and other support staff?) create and add short clips that capture/represent the student's lived experience (e.g., classroom from the student's perspective; how the student has to navigate digital texts, etc.)

### 4.3.5. Share with the Campus Community

The final step of the process is to share the product with the intended audience. The director of the center/partnership program can share the video with all faculty, staff

members, and students, and also perhaps schedule presentations with relevant offices on campus, such as Access Services. A few days after Morgan's video was shared with the W&M community, a student in one of her classes said she had shared the video with a friend of hers who is in high school and has the same vision condition as Morgan. Morgan and that student are now in touch, and Morgan is excited to be a dialogue partner for her as she navigates—and perhaps contributes to transforming—accommodation culture. This is one example of the kind of student-student connection that Jada had anticipated might result from this partnership project.

## 5. Recommendations for Transforming Accommodation Culture into Equity Culture

In this section, we synthesize learnings from the partnership experience we share above to offer recommendations for transforming accommodation culture into equity culture. We embrace Goodley and colleagues' assertion that "disability is both a signifier of inequity and the promise of something new and affirmative" [5]. We also acknowledge Dolmage's warning against checklists that fail to embrace the "patterning of engagement and effort" [16] necessary to create truly accessible and equitable spaces. The guiding principles we offer below, several of which are accompanied by specific examples of how to enact them, might, we hope, help faculty, support staff, and students better understand what some students experience, and ultimately shift the culture toward equity and inclusion for all students. These recommendations are in keeping with Dolmage's reminder that UD calls for "seeing space as multiple and in-process" and recognizing that "we are all involved in the continued production of space (and that students should be agents in this negotiation)" [16].

### 5.1. Establish Diversity as the Norm in Every Learning Context

Rather than assume that there is a single 'normal' way to do anything—a manifestation of what Dolmage has called "academic ableism" [3,16]—assume that there are multiple, legitimate ways that could benefit everyone, not only the people who use them out of necessity. In order to establish this norm, every invitation must, at a minimum, adhere to UD principles and, ideally, recognize that is only a starting place. No approach will work for everyone, so that itself should be the underlying assumption. As Morgan notes in her video, "vision is a spectrum"; this is true across disabilities. As Morgan also notes in the video, each individual intersection of capacities and challenges manifests differently to the observer—and is experienced differently by the one at the intersection.

### 5.2. Intentionally Invite a Revision of Differences from Deficits to Resources

Shifting conceptions of differences from deficits to resources is a multistep process. It starts with the establishment of diversity as the norm noted above, and it entails explicitly asking students to name the dimensions of their diversity that they feel are relevant to their learning—to create structures and conversations focused on such naming. It also means inviting students to conceptualize these differences, their own and others', as resources—a conceptualization that goes against accommodation culture, which continues to assume a narrow uniformity as the norm.

One approach Alison takes in her courses to intentionally support this revision is to ask students to complete what she calls an Access Needs Form, which invites students to: (1) identify any differences or disabilities they think might impact their engagement in the course; (2) specify what they need to support their learning; (3) explain how these differences or disabilities might be resources to themselves and others in the course; and (4) state whether and how they want this information to be shared with the class [20]. This form does not replace the invitations from access services offices; it does, however, enact and call for a culture shift. Taking the time early in a course to reframe differences as resources creates a classroom culture that is more inclusive and equitable. Everyone involved contributes to the co-creation of the learning environment in at least three ways. First, raised awareness informs students' attitudes and quality of attention. Second, students

make deliberate choices to embrace ways of engaging that are inclusive and equitable. Finally, students seek out and support differences that others bring to the course, which deepen understanding [20].

### 5.3. Go beyond Providing Accommodations to Understand Students' Lived Experiences

Offices of access services are often understaffed and overstretched, and they have to navigate legally binding legislation (which requires ways of naming that are still deficit-based and carry potential legal consequences), faculty attitudes (which are sometimes resistant to accommodation, let alone the shift to equity), and students' expressed needs (which are diverse and sometimes shifting). The policies and practices that emerge on campuses that are still working according to accommodation culture principles do not address the complexity of this intersection. As Schley and Marchetti [17] argue:

> While the ADA guarantees basic access to information exchanged in classroom settings and for students demonstrating their knowledge, it does not address a complex set of interaction and collaboration dynamics that occur frequently in higher education classrooms. Only when we remove barriers from the classroom can students experience full participation through inclusion. This requires making space for faculty and students to explore the barriers and friction points and work together to find solutions.

Building on Schley and Marchetti's point, we suggest that this shift is one of both mindset and practice that can be catalyzed by gaining a deeper understanding of students' experiences, capacities, and needs, so those do not remain abstract concepts to faculty and professional staff and to other students. As Gibson and her co-authors argue, efforts to move toward equity culture need to do the following: prioritize 'knowing' our students; understand and value students' individual and collective positions; know students' stories and position their knowledge as powerful (i.e., they are the expert); move institutional culture, practices, and professional knowledge beyond a pejorative/ableist position on 'disability'; and engage meaningfully with disabled students [21]. Resources such as the Human Stories section of Access Insights gathers students' stories in a single place [22].

### 5.4. Share the Active Taking of Responsibility for Shifting from Accommodation to Equity Culture

As undergraduate Isabel Martin notes, we need to move beyond institutions' tendencies to be reactive rather than proactive [23]. Such a move could contribute to a shift from accommodation culture to equity culture. Proactive approaches to actively taking responsibility for such a shift can unfold within and beyond courses.

Within courses, consider how to co-create a classroom environment that takes diversity and differences as the norm and invites students to consider—anonymously or not, depending on the classroom—who is in the room and how everyone can pursue their learning without being blocked by barriers and the mindset of accommodation that, however well-intentioned, perpetuates inequity. The suggestions in the three sections above—to establish diversity as the norm in every learning context, to intentionally invite a revision of differences from deficits to resources, and to go beyond providing accommodations to understand the student's lived experience—are good starting points. Such a shared approach could be understood as analogous to Bovill's arguments for whole-of-class co-creation in relation to various components of course content [24].

Beyond the classroom, develop projects, such as that described by Brown and colleagues [14], that draw on the expertise of differently positioned and differently abled members of higher education contexts to ensure accessibility while also working toward culture change. For instance, the director of libraries at W&M hired Morgan to review library resources for accessibility, drawing not only on her particular expertise regarding what equitable access might look like for students who are not fully sighted, but also for students who might encounter barriers due to cognitive and other physical differences from the assumed norm. Another example is the virtual reality experience created by Annie Krege, a thesis student at Lafayette College. Annie created an "ideal" Lafayette conveyed

through virtual reality technology [25] as a way to demonstrate the experiences of disabled students on Lafayette's campus. Her research argued for the importance of sharing the lived experiences of Lafayette's disabled community in order to create accountability and change the exclusive nature of the campus toward disabled students. Sharing with the campus community both humanizes the student experience and promotes the opportunity for potential change by the intended audience [26].

Finally, seek ways to partner with teaching and learning centers, as Morgan did with W&M's STLI, in an intentional, respectful process that is informed by student experience and expertise and by the center's particular approach. For instance, W&M's STLI currently focuses on crafting stories—narratives that capture and convey students' lived experiences. As described in detail above, Morgan partnered with W&M's STLI to produce a very short video that recreates and effectively conveys her experiences—a one-time investment that will not only save her time and energy in future, but may also contribute to a culture shift. Through short forms of simulation rather than simply text-based descriptions, this approach makes accessible to those who are not facing barriers to engagement and learning the felt and experienced realities of those who are, and compels people to have a deeper understanding of the diversity of students' lived experience and pursue concrete approaches to making classrooms equitable and inclusive.

## 6. Conclusions

We hope that this project is only one of a number of such projects to capture and share the experiences of a wide diversity of students. Simply abiding by ADA legislation and encouraging faculty to embrace and enact UD principles is not sufficient to create an equitable and inclusive culture for students with disabilities in higher education. Pedagogical partnerships create structures through which disabled students, faculty, support staff, and, in some cases, parents can work together to understand, convey, and create conditions for supporting differences as resources. Naming the norms of accommodation culture as inequitable and exclusive and offering alternatives in both language and practice, as Critical Disabilities Studies does, make way for those in higher education not only to insist on "barrier-free design" [6], but also to position students as those with expertise and agency in the negotiation of the production of the spaces in which they learn [14,16].

Educational cultural systems that underserve and marginalize some individuals—causing suffering in maintaining such a distorting and false framework—hinder all from realizing the identities and freedoms only possible within and through an equitable and inclusive community. The co-creation of an equity culture is therefore not only about alleviating individuals' suffering, but also about changing what education is and can be for all people. The intersection of pedagogical partnership work, child-parent research, and Critical Disability Studies, and the choice to draw on the expertise of students and colleagues in areas such as Film Studies, offer tremendous potential to realize "the promise of something new and affirmative" [5]. Across as well as within institutions, we can strive to transform accommodation culture into equity culture.

**Author Contributions:** Both authors contributed equally to the work. All authors have read and agreed to the published version of the manuscript.

**Funding:** This research received no external funding.

**Institutional Review Board Statement:** Not applicable.

**Informed Consent Statement:** Not applicable.

**Data Availability Statement:** Not applicable.

**Acknowledgments:** Thanks to Ryan Harp and Jada Bonds, the student partners who co-created this project with Morgan, and to Mike Blum, the assistant director of the Studio for Teaching and Learning Innovation at The College of William & Mary. Thanks, too, to Kristin Lindgren for suggesting resources and scholarship. Finally, thanks to student and faculty colleagues, Annie Krege, Isabel Martin, Kelly Matthews, and Sarah Slates, for commenting on drafts of the chapter.

**Conflicts of Interest:** The authors declare no conflict of interest.

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
