# Peer review of "From Reporting to Removing Barriers: Toward Transforming Accommodation Culture into Equity Culture"

_education, doi:10.3390/educsci13060611_

Round 1

Reviewer 1 Report

I welcome this paper for its transparency and clarity.

The paper begins with references to established texts/ research.  I suggest that these are revisited in the conclusions.  In addition, it would be useful for the authors to reference legislation and Office for Students on existing guidance - again in the conclusion.

Author Response

Reviewer comment: I welcome this paper for its transparency and clarity.

Response: Thank you for your affirmation of our work.

Reviewer comment: The paper begins with references to established texts/ research.  I suggest that these are revisited in the conclusions.  In addition, it would be useful for the authors to reference legislation and Office for Students on existing guidance - again in the conclusion.

Response: We have added reference to legislation in the Introduction. Here is the revised version:

Both legislation and practice have expanded access in higher education in the United States for students with disabilities. The Americans with Disabilities Act (ADA), passed in 1990, prohibits discrimination against and requires reasonable accommodations for people with disabilities (U.S. Department of Labor, n.d.). However, disability services offices established to meet both legal and ethical obligations and Universal Design (UD) principles created to provide flexible and responsive learning environments offer uneven support for disabled students across US institutions of higher education, with a high percentage of academic staff both unaware of and resistant to UD practices in particular (Gould, Harris and Mullen 2019). These realities create a culture—a set of values and customs—premised on (often grudging and inadequate) accommodation, rather than on equity.

We also return to this addition and to the main points from the literature we review by revising the first paragraph in the conclusion:

We hope that this project is only one of a number of such projects to capture and share the experiences of a wide diversity of students. Simply abiding by ADA legislation and encourage academic staff to embrace and enact UD principles is not sufficient to create an equitable and inclusive culture for students with disabilities in higher education. Pedagogical partnerships create structures through which disabled students, academic and support staff, and, in some cases, parents can work together to understand, convey, and create conditions for supporting differences as resources. Naming the norms of accommodation culture as inequitable and exclusive and offering alternatives in both language and practice, as Critical Disabilities Studies does, make way for those in higher education not only to insist on “barrier-free design” (Hamraie, 2013, para 1) but also to position students as those with expertise and agency in the negotiation of the production of spaces in which they learn (Brown et al. 2020; Dolmage, 2015).

We are not sure what “Office for Students” refers to.

Reviewer 2 Report

This is a very interesting and intriguing paper. I enjoined reading it. However, I do have some suggestions for amelioration the authors can consider.

1.     The paper wants to achieve an equity culture instead of accommodation culture. I think there needs to be more explanation as to what this idea of ‘culture’ might mean and what accommodation and equity culture respectively mean. You only slightly touch these ideas while it is very fundamental for this paper. As to the first, how does this relate to the bigger picture of broader society and politics? Especially when drawing on a CDS perspective, I think it is important to contextualize and explain.

2.     A second idea concerns the double relationship of mother-staff and daughter-student. I think it is very interesting this double perspective is taken into account, but you do not say a lot about it. While I think the structures, habits, creative solutions and contra ableist movements often arise from home. It is not very elaborated on, while this could be very valuable for your objective. Last week, I met a mother and her daughter is a student also, as a mother she is not allowed to enter the campus and support her daughter. At home she has to do this all the time. These perspectives need more attention.

3.     In the first part of the paper it often felt too condense. I think it would be stronger to make some issues more concrete. E.g., line 95, co-create inclusive practices and an equity culture – but what does this mean? How can we make this more tangible?

4.     The text would be more clear if you would add more meta text – introduce what will follow – now I often had the feeling I needed more information and then you come back to it later, but it is not announced before that you will do that.

5.     You talk about access as a very important idea and concept, you connect it to a cultural system (you refer to this a lot, that is why I think it is really necessary to explain more fully what you mean by that), to me it is not clear how you approach access – what is this to you?

6.     I think it will relate to point 1, but how can these ableist norms also be very hard for the other students? Now I sometimes get the feeling this is not really acknowledged, while there is criticism to the whole system?

7.     I liked the personal narrative of author 2 – it is a pity we cannot see the video material. However I also think it would be valuable to refer back to what is seen in the video further on in the article, so the connection between those parts is made more explicit.

8.     Line 232 – You give Guidelines for co-creating representations of students’ lived experiences – I do not see on what this method is based. I think it is important to refer and to make this more explicit.

9.     Line 343 – I would place this other example in the discussion part – I do think it is extremely important to focus on this community idea. VR would better be written full the first time.

10.     Line 425 – you state that the mindset of accommodation perpetuates inequity – do you suggest this whole ‘culture’ should be erased? Or what are you pleading for? And can you make more concrete how this equity culture might look like then without accommodations?

11.     I love the focus on difference and affirmation of difference. I think it would be valuable to make also this idea more concrete – how can we see this in the examples author 2 shares? Is it possible to share this and make more clear how this equity culture starting from affirming difference looks like? What is shared now especially refers back to what was hard when starting to study and having to get the accessible exams as you had requested before. In the latter part of the manuscript there is a lot of focus on the steps for the project, but I miss how this is received or is made concrete or how the ideas you introduce are guarded or implemented there.

I hope these suggestions/questions can be of help for your work.

Author Response

Reviewer comment: The paper wants to achieve an equity culture instead of accommodation culture. I think there needs to be more explanation as to what this idea of ‘culture’ might mean and what accommodation and equity culture respectively mean. You only slightly touch these ideas while it is very fundamental for this paper. As to the first, how does this relate to the bigger picture of broader society and politics? Especially when drawing on a CDS perspective, I think it is important to contextualize and explain.

Response: We have unpacked the admittedly telegraphic introductory points to try to be more explicit. There is insufficient space to go into a discussion of broader society and politics, but we added reference to legislation and practice in the US context, more explicitly defined accommodation culture, and named what equity culture could look like. See revisions to first two paragraphs of the Introduction (lines 25-103).

Reviewer comment: A second idea concerns the double relationship of mother-staff and daughter-student. I think it is very interesting this double perspective is taken into account, but you do not say a lot about it. While I think the structures, habits, creative solutions and contra ableist movements often arise from home. It is not very elaborated on, while this could be very valuable for your objective. Last week, I met a mother and her daughter is a student also, as a mother she is not allowed to enter the campus and support her daughter. At home she has to do this all the time. These perspectives need more attention.

Response: It is true that we do not go into detail about this, again for reasons of space and for wanting to focus on the project at hand. We have added a few more points, though, to try to bring out the important possibilities and barriers you name. These are in the Conceptual Frameworks section (lines 253-258).

Response: Reviewer comment: In the first part of the paper it often felt too condense. I think it would be stronger to make some issues more concrete. E.g., line 95, co-create inclusive practices and an equity culture – but what does this mean? How can we make this more tangible?

Response: We have endeavored to explain this in the Introduction. We have reorganized the Introduction to be more explicit (lines 25-103).

Reviewer comment: The text would be more clear if you would add more meta text – introduce what will follow – now I often had the feeling I needed more information and then you come back to it later, but it is not announced before that you will do that.

Response: We have added more meta text. As noted above, we have reorganized the Introduction so that it more explicitly narrates, in order, what each section aims to achieve. In particular, see lines 124-136.

Reviewer comment: You talk about access as a very important idea and concept, you connect it to a cultural system (you refer to this a lot, that is why I think it is really necessary to explain more fully what you mean by that), to me it is not clear how you approach access – what is this to you?

Response: We offer a more explicit point about access at the beginning of the fifth paragraph of the Introduction—lines 115-117.

Reviewer comment: I think it will relate to point 1, but how can these ableist norms also be very hard for the other students? Now I sometimes get the feeling this is not really acknowledged, while there is criticism to the whole system?

Response: This is an excellent point. Thank you for raising it. We have added a point along these lines to the Conclusion (lines 815-819).

Reviewer comment: I liked the personal narrative of author 2 – it is a pity we cannot see the video material. However I also think it would be valuable to refer back to what is seen in the video further on in the article, so the connection between those parts is made more explicit.

Response: We removed the video for anonymized review, or our identities would have become clear. The editor has indicated that we can de-anonymize, since the paper has been accepted pending minor revisions, so the link is there now. In our discussion, we do reference points raised in the video, but we have made that referencing more detailed (e.g., lines 591-593, 601-604, 672-675).

Reviewer comment: Line 232 – You give Guidelines for co-creating representations of students’ lived experiences – I do not see on what this method is based. I think it is important to refer and to make this more explicit.

Response: This method is based on our own experience, as stated. We have tried to make this clearer (lines 130-133, 466-468).

Reviewer comment: Line 343 – I would place this other example in the discussion part – I do think it is extremely important to focus on this community idea. VR would better be written full the first time.

Response: We have moved this example to the Discussion (lines 782-790).

Reviewer comment: Line 425 – you state that the mindset of accommodation perpetuates inequity – do you suggest this whole ‘culture’ should be erased? Or what are you pleading for? And can you make more concrete how this equity culture might look like then without accommodations?

Response: We have attempted to further clarify. As we assert at several points in the manuscript, it is a matter of replacing the notion and practice of ‘accommodations’ with the premise of difference as the norm and the practice of creating structures that support students engaging in a wide variety of ways.

Reviewer comment: I love the focus on difference and affirmation of difference. I think it would be valuable to make also this idea more concrete – how can we see this in the examples author 2 shares? Is it possible to share this and make more clear how this equity culture starting from affirming difference looks like? What is shared now especially refers back to what was hard when starting to study and having to get the accessible exams as you had requested before. In the latter part of the manuscript there is a lot of focus on the steps for the project, but I miss how this is received or is made concrete or how the ideas you introduce are guarded or implemented there.

Response: Thank you. We have tried to make these connections more explicit. To do so, we have revised the Abstract to be clearer about the focus of the case study, the fifth paragraph of the Introduction, and the opening sentences in sections throughout (all highlighted with Track Changes).

Reviewer comment: I hope these suggestions/questions can be of help for your work.

Response: Yes, they have been very helpful. Thank you.